# SurfSplat: Conquering Feedforward 2D Gaussian Splatting with Surface Continuity Priors

**Bing He [1] & Jingnan Gao [1] & Yunuo Chen [1]
& Gang Chen [2] & Ning Cao [2] & Zhengxue Cheng [1]
& Li Song [1] & Wenjun Zhang [1]**

[1] Shanghai Jiao Tong University,
[2] Tianyi Shilian Technology Co., Ltd
{sandwich_theorem}@sjtu.edu.cn

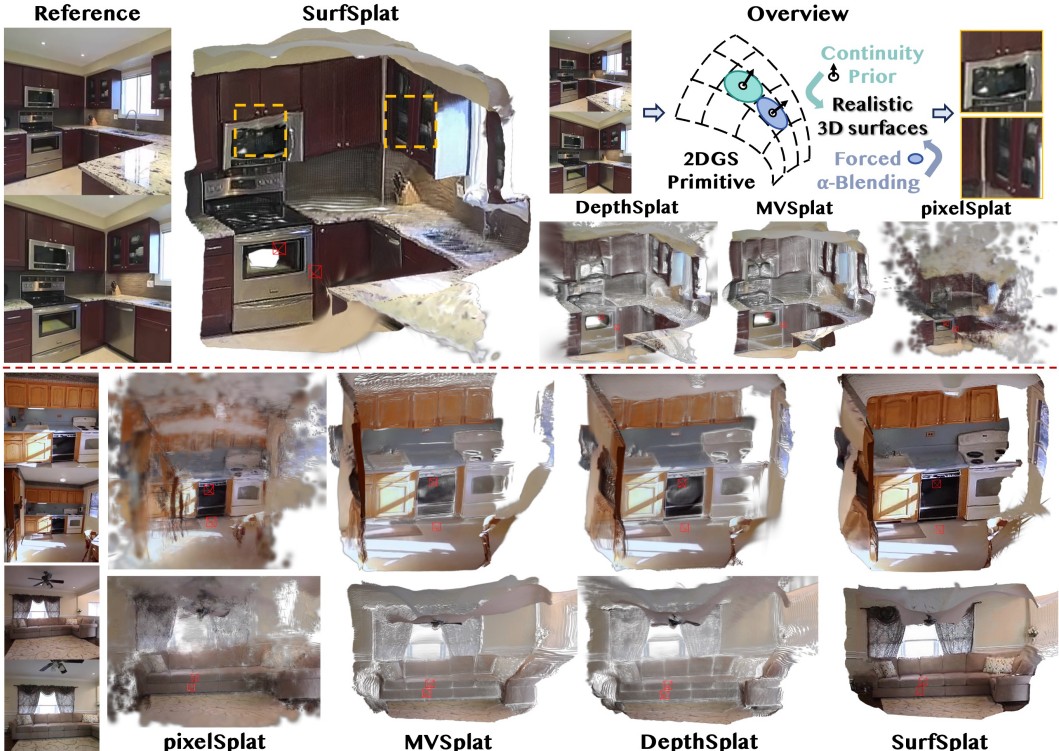

Figure 1: **SurfSplat** is a feedforward network that predicts a 3D scene representation from sparse images input. Previous methods often produce sparse, color-biased pointclouds that lack surface continuity, especially under close-up views. In contrast, our SurfSplat approach utilizes 2DGS with a surface continuity prior and forced alpha blending to generate coherent and realistic 3D surfaces.

## Abstract

Reconstructing 3D scenes from sparse images remains a challenging task due to the difficulty of recovering accurate geometry and texture without optimization. Recent approaches leverage generalizable models to generate 3D scenes using 3D Gaussian Splatting (3DGS) primitive. However, they often fail to produce continuous surfaces and instead yield discrete, color-biased point clouds that appear plausible at normal resolution but reveal severe artifacts under close-up views. To address this issue, we present SurfSplat, a feedforward framework based on 2D Gaussian Splatting (2DGS) primitive, which provides stronger anisotropy and higher geometric precision. By incorporating a surface continuity prior and a forced alpha blending strategy, SurfSplat reconstructs coherent geometry together with faithful textures. Furthermore, we introduce High-Resolution Rendering Consistency (HRRC), a new evaluation metric designed to evaluate high-resolution reconstruction quality. Extensive experiments on RealEstate10K, DL3DV, and ScanNet demonstrate that SurfSplat consistently outperforms prior methods on both standard metrics and HRRC, establishing a robust solution for high-fidelity

3D reconstruction from sparse inputs. Project page: `https://hebing-sjtu.github.io/SurfSplat-website/`

# 1 INTRODUCTION

Reconstructing geometrically accurate real-world scenes continues to be a longstanding challenge in 3D vision. Such capability is crucial for applications like immersive VR experiences, realistic gaming environments, and digital content creation, where both geometric fidelity and visual consistency are essential. To address this, 3D Gaussian Splatting (3DGS) Kerbl et al. (2023) has recently shown impressive performance in novel view synthesis and scene reconstruction. It represents a scene as a collection of discrete, semi-transparent ellipsoids, which are rendered onto the image plane through "splatting". Existing Gaussian-based reconstruction methods mainly follow two paradigms. Traditional approaches, such as vanilla 3DGS, rely on a preprocessing step using COLMAP Schönberger & Frahm (2016) to generate an initial point cloud and typically require access to hundreds of posed views. These methods then perform scene-specific optimization over tens of thousands of iterations, often taking several hours to converge to high-quality results. In contrast, feedforward approaches employ pretrained models to directly predict per-pixel 3D Gaussians from sparse inputs—often as few as two images—without any preprocessing. These methods can reconstruct a 3D scene within milliseconds, enabling real-time and scalable applications.

However, we observe that existing feedforward methods tend to generate degraded 3D scenes. The reconstructed surfaces often collapse into nearly spherical, discrete point clouds with color biases and visible voids. This degradation stems from the under-utilization of the anisotropic properties of Gaussian primitives, which makes it difficult to disentangle geometry from appearance. Moreover, since current feedforward methods rely primarily on image loss, they often yield biased geometry and appearance under sparse or weakly constrained viewpoints. These issues are often subtle in rendered images at the original resolution and near reference views, but become prominent when the camera moves closer or shifts to off-axis viewpoints. This discrepancy indicates that standard novel view synthesis (NVS) metrics fail to accurately capture the geometric and textural fidelity of the scene.

To address these challenges and provide a more accurate reconstruction, we propose SurfSplat, a feedforward model that reconstructs 3D scenes from sparse images using 2D Gaussian Splatting (2DGS) as the representation primitive. Unlike 3DGS, 2DGS captures anisotropic structures more effectively, resulting in improved geometric precision. However, direct training of 2DGS often suffers from instability that arises from the complex coupling between geometric attributes and rendering outcomes. This issue is amplified under limited supervision, where gradients cannot effectively disentangling geometry from appearance. The faceted nature of 2D Gaussians further intensifies the problem, as even minor geometric perturbations can produce substantial deviations in rendered outputs. To tackle this, we introduce two key components: (1) *an explicit surface continuity prior*, which binds the rotation and scale attributes of each 2DGS to its spatial position, encouraging smooth and coherent surfaces. (2) *a forced alpha blending strategy*, which helps the model escape local optima and reduces color bias during training.

Evaluating the quality of 3D scenes produced by feedforward models is also nontrivial. Traditional geometry metrics such as Chamfer Distance or F1 Score are ineffective due to incomplete or sparse outputs and the lack of dense ground truth. Furthermore, most datasets lack out-of-distribution viewpoints for reliable assessment. To address this, we propose **High-Resolution Rendering Consistency (HRRC)**: a novel metric that evaluates scene fidelity by rendering the 3D model at high resolutions, thereby simulating close-up views that expose hidden artifacts like spatial voids. Moreover, HRRC can be computed directly from standard datasets without requiring new annotations.

Built upon these components, SurfSplat reconstructs continuous, high-fidelity 3D scenes with significantly fewer holes and artifacts when viewed from challenging perspectives. Unlike previous 3DGS-based methods that predict Gaussian attributes independently, our approach explicitly models continuity and structure, enhancing both geometric accuracy and rendering consistency.

In summary, the main contributions of this work are as follows:

- We propose **SurfSplat**, a feedforward network that reconstructs 3D scenes using 2D Gaussian surfels from sparse inputs. Our model leverages a surface continuity prior and forced alpha blending to significantly improve reconstruction quality.

- We introduce **HRRC**, a high-resolution rendering-based metric that reveals surface discontinuities and enables fairer evaluation of forward-generated scenes through dense sampling.

- Extensive experiments demonstrate that SurfSplat achieves **state-of-the-art** performance in both standard and HRRC metrics on RealEstate10K, DL3DV, and ScanNet, setting a new benchmark for novel view synthesis under sparse-view settings.

## 2 RELATED WORKS

### 2.1 3D GAUSSIAN SPLATTING

Recent Neural Radiance Field (NeRF) Mildenhall et al. (2021) approach has proven effective for scene reconstruction by leveraging a continuous implicit representation of the scene. Subsequent works have improved reconstruction quality by evolving from MLPs to grid-based structures. For instance, Müller et al. (2022) introduced the Instant Neural Graphics Primitives (Instant-NGP), while Fridovich-Keil et al. (2022) proposed Plenoxels. Other methods, such as Mip-NeRF Barron et al. (2021; 2022), model rays as cones to achieve anti-aliasing.

To accelerate rendering, various strategies have been explored, including precomputation Wang et al. (2023; 2022); Fridovich-Keil et al. (2022); Yu et al. (2021) and hash-based encoding Müller et al. (2022); Takikawa et al. (2022). Additionally, several extensions have adapted NeRF to dynamic scenes Xian et al. (2021); Park et al. (2021a;b); Pumarola et al. (2021); Song et al. (2023).

More recently, 3D Gaussian Splatting (3DGS) Kerbl et al. (2023) introduced an efficient, point-based rendering approach. By representing scenes as collections of semi-transparent, anisotropic Gaussians in 3D space, 3DGS enables photorealistic rendering via rasterization-based splatting.

Numerous extensions have emerged to enhance the capabilities of 3DGS, targeting various aspects such as: optimization efficiency Cheng et al. (2024); Zhang et al. (2024); Radl et al. (2024); Diolatzis et al. (2024), anti-aliasing Yan et al. (2024); Yu et al. (2024); Song et al. (2024); Liang et al. (2024), geometric fidelity Huang et al. (2024), and representation compression for faster inference Girish et al. (2024); Navaneet et al. (2024); Niedermayr et al. (2024); Lee et al. (2024); Fan et al. (2024); Chen et al. (2024a). Efforts to extend 3DGS to dynamic scenes have also been explored Luiten et al. (2023); Wu et al. (2023); Wan et al. (2024); Huang et al. (2023); He et al. (2024).

Among these, Huang et al. (2024) proposed 2DGS, a novel differentiable surface element capable of representing surfaces with higher accuracy. However, conventional 3DGS pipelines typically require precomputed sparse point clouds, accurate camera poses, and extensive per-scene optimization, limiting their applicability in sparse-view settings.

### 2.2 GENERALIZABLE 3D RECONSTRUCTION

To alleviate the need for costly per-scene optimization, recent works explored feedforward networks that directly predict 3D Gaussians from sparse image collections.

Splatter image Szymanowicz et al. (2024) proposed a novel paradigm for converting images into Gaussian attribute images. Other approaches incorporated task-specific backbones to improve reconstruction by leveraging geometric cues. For example, PixelSplat Charatan et al. (2024) used epipolar geometry for efficient depth estimation, while MVSplat Chen et al. (2024b) builded cost volumes to aggregate multi-view information.

Follow-up works further extended these ideas. FreeSplat Wang et al. (2024b) addressed limited synthesis range via a pixel-wise triplet fusion strategy. Hisplat Tang et al. (2024) predicted multiple Gaussian layers in a hierarchical structure. DepthSplat Xu et al. (2024b) enabled cross-task interaction between depth estimation and Gaussian splatting.

Several researches also focused on improving generalization by introducing triplane representations Zou et al. (2024); Xu et al. (2024a). SplatFormer Chen et al. (2024d) leveraged pretrained models to improve performance in out-of-distribution views. NopoSplat Ye et al. (2024) abandoned the transform-then-fuse pipeline and directly generated 3D scenes in canonical space. G3R Chen et al. (2024c) extended the generalizable 3DGS to dynamic scenes using auxiliary LiDAR data.

Despite these advancements, prior feedforward methods primarily rely on 3DGS primitives. Without effective regularization, the generated 3D scenes often lack realistic and continuous surfaces. These degradations are typically unseen at original resolution near reference views, but become apparent under close-up or off-axis inspection.

In contrast, our approach adopts 2DGS as the scene representation primitive. By introducing a surface continuity prior and a forced alpha blending technique, our model successfully trains highly anisotropic surface elements, enabling high-fidelity 3D scene reconstruction from sparse inputs.

## 3 METHOD

### 3.1 PRELIMINARIES

Feedforward 3D Gaussian Splatting (3DGS) methods aim to regress a set of 3D Gaussians directly from sparse multi-view images. Unlike optimization-based approaches that iteratively refine Gaussians, feedforward methods predict all Gaussian parameters in a single forward pass. Given a collection of $V$ input images $\{I^v\}_{v=1}^V$ with corresponding camera intrinsics $\{\mathbf{k}^v\}_{v=1}^V$ and poses $\{\mathbf{T}^v\}_{v=1}^V$, the network $f_\theta$ predicts Gaussian parameters for each pixel as:

$$f_\theta : \{(I^v, \mathbf{k}^v, \mathbf{T}^v)\}_{v=1}^V \mapsto \left\{ \bigcup_{j=1}^{H \times W} \left( \boldsymbol{\mu}_j^v, \boldsymbol{\alpha}_j^v, \mathbf{r}_j^v, \mathbf{s}_j^v, \mathbf{c}_j^v \right) \right\}_{v=1}^V, \tag{1}$$

where $\boldsymbol{\mu}_j^v$ denotes the 3D position, $\boldsymbol{\alpha}_j^v$ the opacity, $\mathbf{r}_j^v$ the rotation, $\mathbf{s}_j^v$ the scale, and $\mathbf{c}_j^v$ the spherical harmonics of the $j$-th Gaussian generated from the $v$-th view. The feasibility of such models arises from the observation that, even with sparse-view conditions, image features extracted by modern backbones (e.g., ViTs Ranftl et al. (2021); Zhang et al. (2022); Wang et al. (2024a)) retain sufficient local geometric cues for direct 3D reasoning. When combined with the camera intrinsics, these features can be projected into 3D space and assigned accurate Gaussian attributes, enabling end-to-end training via differentiable rasterization and photometric reconstruction loss.

### 3.2 MODEL ARCHITECTURE

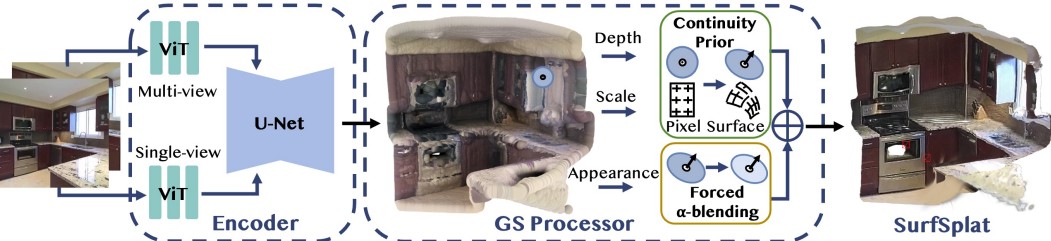

Figure 2: **Illustration for model architecture.** Given sparse input images, our dual-path encoder processes them through both single-view and multi-view branches. The fused features are passed through a U-Net to predict intermediate attributes, including depth, scale multipliers, and appearance components. Finally, these intermediates are converted into standard Gaussian attributes using our surface continuity prior and forced alpha blending strategy.

In the context of feedforward 3D Gaussian Splatting (3DGS), multi-view cues are essential for enforcing geometric consistency across views, while single-view priors offer guidance in regions with missing textures or insufficient correspondences. To integrate these complementary sources effectively, we adopt a dual-path for feature extraction within our model architecture. In the **single-view branch**, we leverage a pretrained monocular depth backbone. Specifically, we use the Depth Anything V2 model Yang et al. (2024), and bilinearly upsample its output features to the target spatial resolution. In the **multi-view branch**, input images are first converted into low-resolution feature maps, which are then processed by multiple layers of self- and cross-attention Vaswani et al. (2017); Liu et al. (2021b) to extract inter-view correspondences. The fused features are subsequently used to construct cost volumes Chen et al. (2024b) across views via the plane-sweep stereo approach Collins (1996); Xu et al. (2023), which serve as the output of the multi-view branch. The final feature representation is obtained by concatenating the single-view and multi-view features.

The combined feature is fed into a 2D U-Net Ronneberger et al. (2015); Rombach et al. (2022) to regress the Gaussian Splatting (GS) attributes, including depth, scale multipliers, higher-order spherical harmonics components, and opacity. These outputs are upsampled to full resolution using a DPT head Ranftl et al. (2021) and further processed with our surface continuity prior and forced alpha-blending techniques to produce the final standard Gaussian attributes. Technical details are provided in Appendix A.1.

### 3.3 SURFACE CONTINUITY PRIOR

Existing feedforward 3DGS methods often produce incoherent and discontinuous surfaces. This stems from the fact that learnable Gaussian primitives struggle to decouple geometry and texture attributes

when trained solely through gradient-based supervision. A closer inspection of rendered results reveals biased color assignments, surface discontinuities, and voids. While these primitives may collectively produce visually plausible images under common rendering settings, the underlying 3D assets remain structurally flawed and fall short of the fidelity required for high-quality 3D generation. To address these issues, we start by an observation: **most visible geometry in real-world scenes consists of smooth, continuous surfaces**. This motivates the introduction of a *surface continuity prior*, which assumes that spatially adjacent surfels on a coherent 3D surface generally correspond to neighboring pixels in the image. Guided by this prior, Gaussians are expected to exhibit correlated geometric attributes. Specifically, the rotation and scale of each Gaussian should be strongly aligned with the positions of its neighboring Gaussians. We consider the image-space neighborhood around a pixel at $(h, w)$, whose associated Gaussian has a 3D position $\mathbf{p}_0 \in \mathbb{R}^3$, with neighboring positions $\{\mathbf{p}_i\}_{i=1}^k$. Following the standard COLMAP coordinate convention, where the camera frame has $x$ pointing right, $y$ downward, and $z$ inward, we assume that the default (unrotated) surface normal aligns with the canonical vector $\mathbf{n}_0 = (0, 0, 1)^\top$. The initial rotation $\mathbf{R}_0 \in SO(3)$ is set to the identity matrix, which corresponds to the quaternion $(1, 0, 0, 0)$.

To estimate the local surface orientation, we apply rightward and downward Sobel filters over the $3 \times 3$ neighborhood around $\mathbf{p}_0$, obtaining two virtual neighbors, $\mathbf{p}_1$ and $\mathbf{p}_2$. These neighbors define two tangent vectors:

$$\mathbf{t}_1, \mathbf{t}_2 = \mathbf{p}_1 - \mathbf{p}_0, \quad \mathbf{p}_2 - \mathbf{p}_0. \tag{2}$$

Although $\mathbf{t}_1$ and $\mathbf{t}_2$ are not guaranteed to be orthogonal in world space, their projections onto the image plane are orthogonal. The local surface normal $\mathbf{n} \in \mathbb{R}^3$ is then computed as their cross product:

$$\mathbf{n} = \frac{\mathbf{t}_1 \times \mathbf{t}_2}{\|\mathbf{t}_1 \times \mathbf{t}_2\|}. \tag{3}$$

Given this target normal $\mathbf{n}$, the corresponding rotation matrix $\mathbf{R} \in SO(3)$ that aligns $\mathbf{n}_0$ with $\mathbf{n}$ can be computed using Rodrigues' rotation formula:

$$\mathbf{R} = \mathbf{I} + [\mathbf{v}]_\times + \frac{1 - c}{\|\mathbf{v}\|^2}[\mathbf{v}]_\times^2, \tag{4}$$

where $\mathbf{v} = \mathbf{n}_0 \times \mathbf{n}$, $c = \mathbf{n}_0^\top \mathbf{n}$, and $[\mathbf{v}]_\times$ denotes the skew-symmetric matrix of $\mathbf{v}$. This rotation aligns the canonical frame with the estimated local surface, giving the updated surfel rotation:

$$\mathbf{R}_{\mathrm{surf}} = \mathbf{R}\mathbf{R}_0 = \mathbf{R}. \tag{5}$$

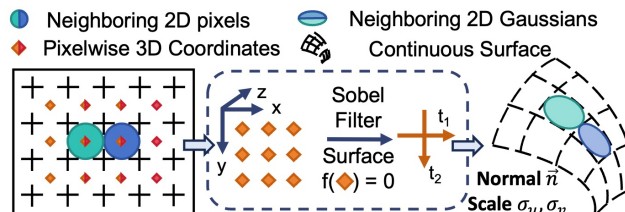

Figure 3: **Illustration for Gaussian processor.** We visualize how image-space neighboring pixels are transformed into Gaussians aligned on a continuous surface via the surface continuity prior. To prevent opacity collapse and preserve 3D alignment, we apply a forced alpha-blending strategy that reduces opacities, ensuring that spatially occluded Gaussians still contribute during rendering.

To define anisotropic scale $\mathbf{S} = \mathrm{diag}(\sigma_u, \sigma_v, \sigma_w)$, we compute the variance of projected neighboring points along the rotated tangent axes $\mathbf{t}_u, \mathbf{t}_v$. Since we employ 2D Gaussian splats, the scale along the depth axis $\sigma_w$ is fixed to zero. To account for screen-space deformation, let $\mathbf{W} \in \mathbb{R}^{4 \times 4}$ denote the transformation matrix from world space to screen space, and let $\mathbf{J}$ represent the Jacobian of the affine approximation of the projective transformation:

$$\Sigma = \mathbf{R}\mathbf{S}\mathbf{S}^\top\mathbf{R}^\top, \tag{6}$$

$$\Sigma' = \mathbf{J}\mathbf{W}\Sigma\mathbf{W}^\top\mathbf{J}^\top, \tag{7}$$

where $\Sigma'$ corresponds to a unit circle in the image plane, as in feedforward methods each GS corresponds one-to-one with an image pixel and its projection always covers a single pixel.

However, **inverting the projection matrix to estimate scale often yields unstable values that hinder convergence.** To address this, we adopt a coarse scale estimate based on image-space distances between neighboring pixels:

$$\bar{\sigma}_u^2, \bar{\sigma}_v^2 = \mathbf{t}_{1x}^2 + \mathbf{t}_{1z}^2, \quad \mathbf{t}_{2y}^2 + \mathbf{t}_{2z}^2. \tag{8}$$

We then use the neural network to predict scale multipliers $\hat{\sigma}_u, \hat{\sigma}_v$, which are constrained to lie within $\left[\frac{1}{3}, 3\right]$. The final scales are then computed as:

$$\sigma_u = \bar{\sigma}_u \hat{\sigma}_u, \quad \sigma_v = \bar{\sigma}_v \hat{\sigma}_v. \tag{9}$$

With this design, instead of directly regressing Gaussian attributes, our method derives them from predicted 3D positions, guided by a physically grounded constraint to ensure spatial consistency. This formulation provides a geometry-aware initialization of 2D Gaussian splats in 3D space, ensuring that their orientation and shape remain consistent with surface continuity.

### 3.4 FORCED ALPHA BLENDING

While the surface continuity prior imposes effective local geometric constraints for continuous 3D reconstruction, we observe that it can lead to suboptimal local minima during training. Specifically, the model tends to learn highly opaque Gaussians, where individual splats saturate the pixel opacity. This behavior rapidly boosts image quality for near-input viewpoints, but under the alpha-blending rendering rule, occluded Gaussians contribute minimally to the output:

$$C = \sum_{i \in \mathcal{N}} c_i \alpha_i \prod_{j=1}^{i-1}(1 - \alpha_j), \quad \alpha = \sum_{i \in \mathcal{N}} \alpha_i \prod_{j=1}^{i-1}(1 - \alpha_j). \tag{10}$$

As a result, deeper Gaussians in the rendering order are effectively ignored, which impairs the model's ability to learn 3D structure and maintain alignment.

To address this, we propose a ***forced alpha blending*** strategy that explicitly constrains each Gaussian's opacity. We clip the predicted opacity using an upper bound $\tau_{\text{opa}} < 1$, ensuring that all Gaussians contribute to the rendering regardless of their depth order. This preserves both the model's multi-layer expressiveness and its 3D alignment capabilities. To further improve the reliability of spherical harmonics (SH)-based color estimation under enforced blending, we apply two adjustments. First, we initialize the RGB color directly into the DC component of the SH basis. Second, We normalize the rendered output $C$ to compensate for transparency, since the final alpha holds $\alpha < 1$ by design:

$$C = \begin{cases} C, & \alpha < \tau_\alpha, \\ \dfrac{C}{\alpha}, & \alpha \geq \tau_\alpha, \end{cases} \tag{11}$$

where $\tau_\alpha$ is a stability threshold to avoid amplifying noise in regions with very low transparency. This correction allows the model to produce unbiased and stable renderings, while maintaining accurate 3D alignment in sparse-view scenarios.

### 3.5 TRAINING LOSS

Our training loss is an image-level loss computed directly between the rendered image and the ground-truth image. We use a combination of mean squared error (MSE) and perceptual similarity (LPIPS):

$$L_{\text{gs}} = \sum_{m=1}^{M} \left( \text{MSE}\left(I_{\text{render}}^m, I_{\text{gt}}^m\right) + \lambda \cdot \text{LPIPS}\left(I_{\text{render}}^m, I_{\text{gt}}^m\right) \right), \tag{12}$$

where $M$ denotes the batch size. The weight $\lambda$ is set to 0.05, following prior works Charatan et al. (2024); Chen et al. (2024b); Xu et al. (2024b).

### 3.6 HIGH-RESOLUTION RENDERING CONSISTENCY (HRRC)

To better evaluate the geometric fidelity of reconstructed 3D scenes, we propose a novel evaluation metric: **High-Resolution Rendering Consistency (HRRC)**.

Conventional metrics—such as PSNR, SSIM, and LPIPS—are typically computed at the same resolution as the input images (e.g., $256 \times 256$). However, these metrics often fail to reveal geometric inaccuracies or sparsity-induced artifacts, which may be hidden at lower resolutions but become apparent under high-frequency sampling.

To address this limitation, we render each reconstructed scene at a higher resolution (e.g., $2\times$ or $4\times$ the original), resulting in an output $\hat{I}^{HR}$. We compare this against a bicubic-upsampled version of the ground truth image, denoted $\hat{I}^{GT\uparrow}$, and compute standard quality metrics:

$$\text{HRRC}_{\text{metric}} = \text{metric}(\hat{I}^{HR}, \hat{I}^{GT\uparrow}) \quad \text{where metric} \in \{\text{PSNR}, \text{SSIM}, \text{LPIPS}\}. \tag{13}$$

HRRC can effectively expose geometric flaws such as sparsity-induced holes, degenerate Gaussian shapes (e.g., overly isotropic splats), and discontinuities in unobserved regions. A higher HRRC score indicates stronger spatial generalization and more accurate 3D reconstruction. This makes HRRC particularly useful for distinguishing models that merely memorize sparse views from those that truly recover 3D geometry.

# 4 EXPERIMENT

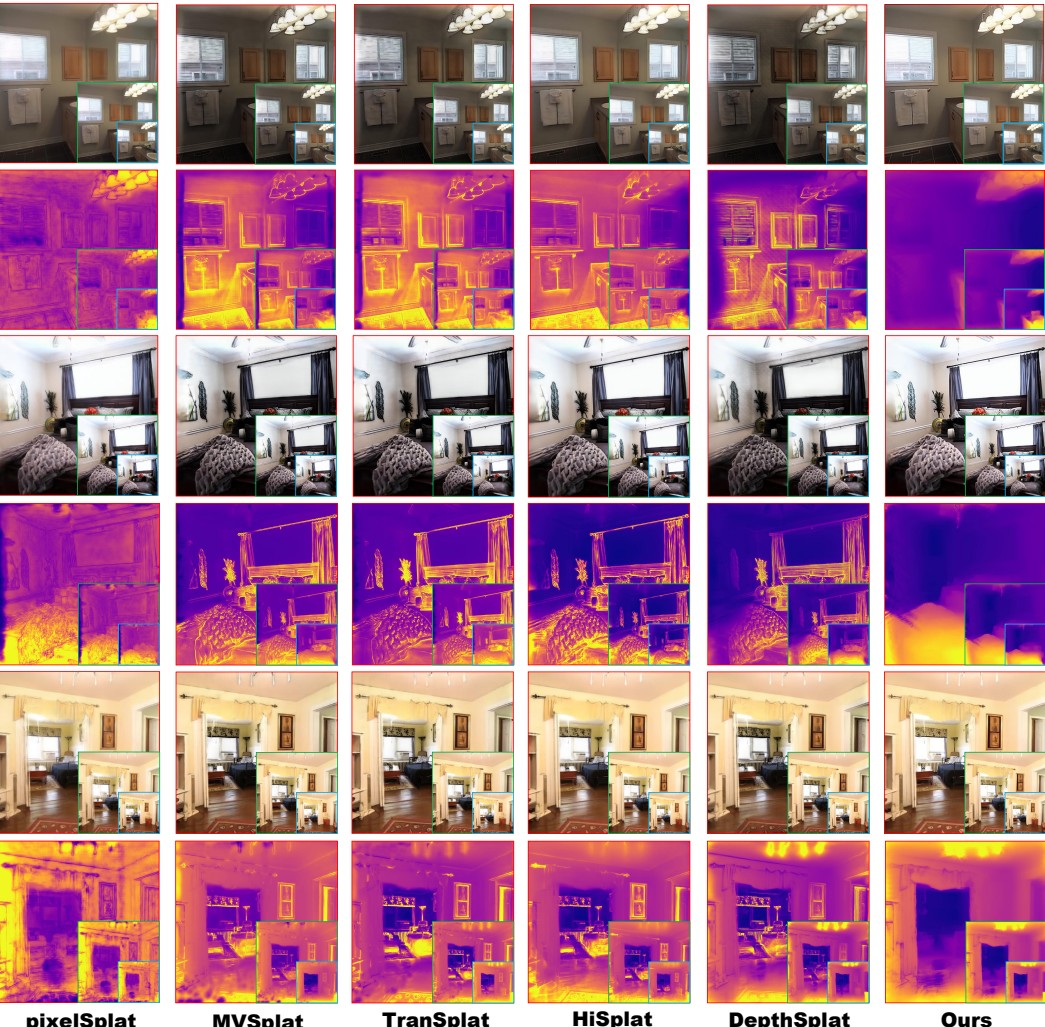

| pixelSplat | MVSplat | TranSplat | HiSplat | DepthSplat | Ours |

Figure 4: **Multi-resolution rendering of 3D scenes.** We visualize rendered images and depth maps at three resolutions: ×1 (blue box), ×2 (green box), and ×4 (red box). As resolution increases, artifacts in the underlying 3D representation become more evident. In the image space, they appear as dark regions caused by unfilled gaps, where hollow areas are rendered as black pixels. In the depth space, they appear as unnatural yellow regions, indicating incorrect depth predictions caused by geometric discontinuities or sparsity. Note that yellow corresponds to near surfaces and blue denotes distant regions in depth map visualization.

**Datasets.** To evaluate our method, we follow the experimental setup in PixelSplat Charatan et al. (2024) and conduct experiments on the RealEstate10K (RE10K) Zhou et al. (2018) and ACID Liu et al. (2021a) datasets. RE10K mainly consists of indoor real estate videos, whereas ACID contains outdoor scenes captured by aerial drones. Both datasets provide precomputed camera poses and we adhere to the official train-test splits used in prior work. Additionally, we evaluate our method on the DTU Jensen et al. (2014) dataset following MVSplat Chen et al. (2024b), on DL3DV Ling et al.

(2024) following DepthSplat Xu et al. (2024b), and further extend our evaluation to the challenging ScanNet Dai et al. (2017) dataset.

**Evaluation Metrics.** We evaluate novel view synthesis quality using standard metrics: PSNR, SSIM, and LPIPS. To better evaluate geometric fidelity, we additionally report high-resolution rendering consistency (HRRC) results at $2\times$ and $4\times$ resolution.

**Baselines.** We compare our method to state-of-the-art sparse-view generalizable methods for novel view synthesis, including PixelSplat Charatan et al. (2024), MVSplat Chen et al. (2024b), TranSplat Zhang et al. (2025), HiSplat Tang et al. (2024), and DepthSplat Xu et al. (2024b). Among these, PixelSplat and HiSplat generate multiple Gaussians per pixel, while MVSplat, TranSplat, and DepthSplat predict a single Gaussian per pixel. Since using more primitives generally improves performance, we focus our core comparisons on the latter group to ensure a fair comparison.

**Implementation Details.** Our method is implemented using PyTorch Paszke et al. (2019) and optimized using AdamW Loshchilov & Hutter (2017) with a cosine learning rate schedule. We conduct experiments with different monocular backbones from Depth Anything V2 Yang et al. (2024) (ViT-S, ViT-B, ViT-L), referred to as Ours-S, Ours-B, and Ours-L respectively. We train our models for a total of 4800K iterations on an NVIDIA A100 GPU following DepthSplat Xu et al. (2024b),. For the small model (Ours-S), we train for 300K iterations with a batch size of 16, while the base and large models (Ours-B and Ours-L) are trained for 600K iterations with a batch size of 8. We adopt the encoder settings from DepthSplat Xu et al. (2024b), but use a lower learning rate of $2 \times 10^{-6}$ for the pretrained Depth Anything V2 backbone. All other layers are trained with a learning rate of $2 \times 10^{-4}$. The opacity threshold $\tau_{\text{opa}}$ is set to 0.6, and the alpha normalization threshold $\tau_\alpha$ is set to 0.1 during training and 0.001 during evaluation. Predicted scale multipliers are clamped to the range $[\frac{1}{3}, 3]$. We train our models at $256 \times 256$ resolution for fair comparison unless otherwise specified. Furthermore, we explore higher-resoluton training at $256 \times 448$ and demonstrate the results in the appendix A.3.

## 4.1 MAIN RESULTS

Table 1: Novel view synthesis performance on the RealEstate10k dataset.

| Method | 256×256 (**Standard**) | | | 512×512 (**HRRC**) | | | 1024×1024 (**HRRC**) | | | Average | | |
|---|---|---|---|---|---|---|---|---|---|---|---|---|
| | PSNR↑ | SSIM↑ | LPIPS↓ | PSNR↑ | SSIM↑ | LPIPS↓ | PSNR↑ | SSIM↑ | LPIPS↓ | PSNR↑ | SSIM↑ | LPIPS↓ |
| pixelSplat | 26.049 | 0.862 | 0.137 | 25.782 | 0.868 | 0.207 | 24.920 | 0.877 | 0.269 | 25.584 | 0.869 | 0.204 |
| HiSplat | 27.193 | 0.882 | 0.117 | 25.269 | 0.870 | 0.198 | 24.262 | 0.878 | 0.248 | 25.575 | 0.877 | 0.188 |
| MVSplat | 26.359 | 0.868 | 0.129 | 20.408 | 0.809 | 0.290 | 17.966 | 0.755 | 0.425 | 21.578 | 0.811 | 0.281 |
| TranSplat | 26.687 | 0.875 | 0.125 | 20.610 | 0.815 | 0.286 | 18.154 | 0.761 | 0.427 | 21.817 | 0.817 | 0.279 |
| DepthSplat | 27.504 | 0.890 | **0.112** | 20.031 | 0.774 | 0.341 | 16.385 | 0.635 | 0.491 | 21.307 | 0.766 | 0.315 |
| **Ours-S** | 27.001 | 0.883 | 0.118 | 25.989 | 0.860 | 0.223 | 24.535 | 0.835 | 0.325 | 25.842 | 0.859 | 0.222 |
| **Ours-B** | 27.447 | 0.890 | 0.113 | 26.280 | **0.866** | 0.218 | 24.744 | 0.838 | 0.322 | 26.157 | 0.865 | 0.217 |
| **Ours-L** | **27.537** | **0.892** | **0.112** | **26.331** | **0.866** | **0.217** | **24.897** | **0.842** | **0.320** | **26.255** | **0.867** | **0.216** |

Table 2: Novel view synthesis performance on the ACID dataset.

| Method | 256×256 (**Standard**) | | | 512×512 (**HRRC**) | | | 1024×1024 (**HRRC**) | | | Average | | |
|---|---|---|---|---|---|---|---|---|---|---|---|---|
| | PSNR↑ | SSIM↑ | LPIPS↓ | PSNR↑ | SSIM↑ | LPIPS↓ | PSNR↑ | SSIM↑ | LPIPS↓ | PSNR↑ | SSIM↑ | LPIPS↓ |
| pixelSplat | 28.284 | 0.842 | 0.146 | 27.687 | 0.848 | 0.243 | 26.462 | 0.858 | 0.343 | 27.478 | 0.849 | 0.244 |
| HiSplat | 28.737 | 0.853 | 0.132 | 25.376 | 0.833 | 0.246 | 23.988 | 0.841 | 0.314 | 25.700 | 0.842 | 0.231 |
| MVSplat | 28.202 | 0.842 | 0.145 | 17.802 | 0.711 | 0.406 | 14.784 | 0.572 | 0.567 | 20.263 | 0.708 | 0.373 |
| TranSplat | **28.337** | **0.845** | **0.143** | 17.911 | 0.716 | 0.402 | 14.956 | 0.582 | 0.558 | 20.401 | 0.714 | 0.373 |
| **Ours** | 28.336 | **0.845** | 0.144 | **26.868** | **0.814** | **0.281** | **21.253** | **0.690** | **0.457** | **25.486** | **0.783** | **0.294** |

**Reconstruction Quality.** We report quantitative comparison on the RE10K dataset in Table 1 and on the ACID dataset in Table 2. Our proposed SurfSplat method consistently outperforms previous state-of-the-art methods across various metrics and datasets, especially under high-resolution rendering settings. As shown in Figure 4, we visualize the predicted 3D scenes rendered into both RGB and depth maps at the original, $\times 2$, and $\times 4$ resolutions. While previous methods appear visually plausible at the original resolution, their reconstructions manifest spatial inconsistencies at higher resolutions, including holes and surface gaps. These artifacts reveal the limitations of previous feedforward 3DGS

Table 3: Cross datasets performance.

| Method | Scannet | | | DL3DV | | | DTU | | | Average | | |
|---|---|---|---|---|---|---|---|---|---|---|---|---|
| | PSNR↑ | SSIM↑ | LPIPS↓ | PSNR↑ | SSIM↑ | LPIPS↓ | PSNR↑ | SSIM↑ | LPIPS↓ | PSNR↑ | SSIM↑ | LPIPS↓ |
| pixelSplat | 19.606 | 0.714 | 0.324 | 27.201 | 0.882 | 0.104 | 12.752 | 0.329 | 0.639 | 19.853 | 0.642 | 0.356 |
| HiSplat | 19.095 | 0.691 | 0.342 | 26.242 | 0.869 | 0.112 | 16.019 | 0.671 | 0.277 | 20.452 | 0.744 | 0.244 |
| MVSplat | 18.725 | 0.692 | 0.333 | 23.841 | 0.768 | 0.156 | 13.914 | 0.470 | 0.386 | 18.827 | 0.643 | 0.292 |
| TranSplat | 18.944 | 0.705 | 0.332 | 23.913 | 0.771 | 0.161 | _14.956_ | **0.527** | **0.327** | 19.271 | 0.668 | _0.273_ |
| DepthSplat | _20.201_ | **0.735** | **0.305** | **28.141** | **0.905** | **0.083** | 14.592 | 0.425 | 0.436 | _20.978_ | _0.688_ | 0.275 |
| **Ours** | **20.305** | _0.731_ | _0.313_ | _27.384_ | _0.890_ | _0.106_ | **15.544** | _0.488_ | _0.329_ | **21.078** | **0.703** | **0.249** |

Table 4: Ablations study on various components.

| Method | 256×256 (**Standard**) | | | 512×512 (**HRRC**) | | | 1024×1024 (**HRRC**) | | | Average | | |
|---|---|---|---|---|---|---|---|---|---|---|---|---|
| | PSNR↑ | SSIM↑ | LPIPS↓ | PSNR↑ | SSIM↑ | LPIPS↓ | PSNR↑ | SSIM↑ | LPIPS↓ | PSNR↑ | SSIM↑ | LPIPS↓ |
| w/o FAB, SCP | 26.925 | 0.880 | 0.120 | 21.549 | 0.805 | 0.307 | 18.563 | 0.716 | 0.422 | 22.346 | 0.800 | 0.283 |
| w/o FAB | 26.481 | 0.873 | 0.128 | 21.042 | 0.776 | 0.345 | 17.576 | 0.662 | 0.474 | 21.700 | 0.770 | 0.316 |
| Full | **27.001** | **0.883** | **0.118** | **25.989** | **0.860** | **0.223** | **24.535** | **0.835** | **0.325** | **25.842** | **0.859** | **0.222** |

models in capturing sub-pixel-level geometry. Notably, DepthSplat, despite using the same encoder backbone as our method, fails to generate coherent geometry or consistent surface details, which highlights the effectiveness of our surface continuity prior and forced alpha blending strategy.

**Cross-Dataset Generalization.** To assess cross-dataset generalization, we train our model on RE10K and directly conduct evaluation on DTU, DL3DV, and ScanNet datasets. As shown in Table 3, SurfSplat maintains strong performance and generalizes better than previous methods across all target domains. This demonstrates the robustness of our learned geometric prior and the general applicability of our representation even under domain shift.

## 4.2 Ablation and Analysis

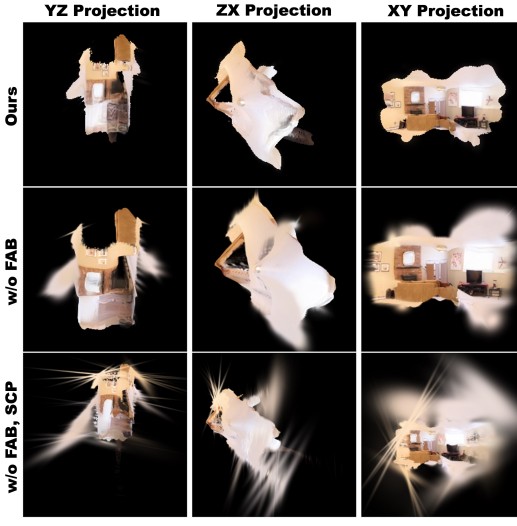

Figure 5: **Ablation study: Visualization of reconstructed 3D scenes.** Our full model yields continuous and coherent surfaces, while ablated variants exhibit visible artifacts and spatial inconsistencies.

We conduct extensive ablation studies to further validate the effectiveness of key components. Specifically, we evaluate variants without both of *forced alpha blending* and *surface continuity prior* (denoted as **w/o FAB,SCP**), and without *forced alpha blending* (denoted as **w/o FAB**). Quantitative results are reported in Table 4, and we also rendered the reconstructed 3D scenes onto three orthogonal planes in Figure 5 to provide qualitative comparisons. Our full model yields continuous and coherent surfaces, while ablated variants exhibit visible artifacts and spatial inconsistencies.

**Surface Continuity Prior.** To evaluate the impact of the surface continuity prior, we train a variant that independently predicts all Gaussian attributes without geometric coupling. Interestingly, this variant still achieves competitive novel view synthesis (NVS) performance at the original resolution, despite producing visually noisy and discontinuous surfaces. This observation highlights a key limitation of conventional NVS metrics and underscores the value of our proposed HRRC metric, which drops significantly when surface continuity is not enforced.

**Forced Alpha Blending.** We also train a variant with the surface continuity prior but without forced alpha blending. We observe a clear spatial misalignment across views, as the model tends to produce fully opaque Gaussians, which occlude background information and hinder correct 3D alignment. This leads to a substantial drop in both standard and HRRC metrics.

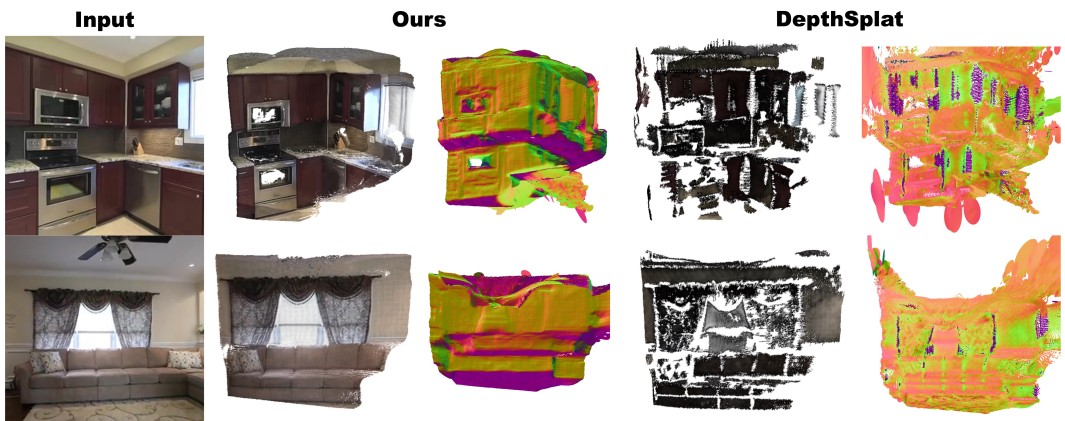

Figure 6: Normal and mesh comparison with DepthSplat.

Table 5: Quantitative performance comparison on high-resolution DL3DV dataset.

| Metric | pixelSplat | HiSplat | MVSplat | TransSplat | DepthSplat | Ours |
|---|---|---|---|---|---|---|
| PSNR ↑ | 24.082 | 22.780 | 17.966 | 19.545 | 16.066 | **24.411** |
| SSIM ↑ | 0.755 | 0.765 | 0.645 | 0.679 | 0.600 | **0.788** |
| LPIPS ↓ | 0.250 | 0.237 | 0.301 | 0.257 | 0.424 | **0.252** |

### 4.3 EFFECTIVENESS OF HRRC METRIC

To empirically validate the effectiveness of HRCC on native high-resolution data, we conducted additional experiments on the high-resolution version of the DL3DV dataset. We randomly sampled a representative subset for evaluation and ensured that all methods were tested under identical conditions. The results are reported in Table 5. Across these experiments, the relative performance rankings remained fully consistent with those observed under HRRC evaluation, even without any bicubic upsampling. This indicates that the conclusions drawn from HRRC reliably transfer to native high-resolution evaluations.

### 4.4 NORMAL AND MESH COMPARISON

Since our method naturally predicts a surface orientation for each 2DGS, we additionally generate the corresponding normal maps and reconstructed meshes to further demonstrate the effectiveness of SurfSplat. We provide a comparison with DepthSplat Yang et al. (2024) in Figure 6. From this comparison, we observe that our method produces more geometrically consistent results, highlighting the improved geometric coherence induced by the surface continuity prior.

## 5 CONCLUSION

We present **SurfSplat**, a feedforward framework for high-fidelity 3D scene reconstruction from sparse views using 2D Gaussian splatting primitive. By introducing a *surface continuity prior* and a *forced alpha blending* strategy, our method addresses key limitations of previous approaches, eliminating surface discontinuities and overcoming opacity collapse. We also propose the **HRRC** metric to better evaluate fine-grained geometric fidelity. Extensive experiments across multiple datasets demonstrate that SurfSplat achieves state-of-the-art performance across both standard and high-resolution metrics, providing a scalable and accurate solution for generalizable 3D reconstruction.

**Limitations.** Despite these improvements, our method still relies on known camera poses, and predicting one Gaussian per pixel can lead to redundant representations. These limitations open opportunities for future research on joint pose elimination and compact, adaptive representations.

## 6 ACKNOWLEDGMENTS AND DISCLOSURE OF FUNDING

This work was partly supported by the NSFC (62431015, 62571317, 62501387), the Fundamental Research Funds for the Central Universities, Shanghai Key Laboratory of Digital Media Processing and Transmission under Grant 22DZ2229005, 111 project BP0719010.

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

# A  TECHNICAL APPENDICES AND SUPPLEMENTARY MATERIAL

## A.1  ENCODER ARCHITECTURE

We adopt a dual-branch encoder design to extract both monocular and multi-view features for robust 3D reasoning, following the architecture proposed by DepthSplat Xu et al. (2024b).

**Multi-view Branch.**  The multi-view encoder begins with a lightweight ResNet-style backbone composed of stride-2 convolutional layers, yielding spatially downsampled feature maps by a factor of $s$. To enable view aggregation, we employ a multi-view Swin Transformer Liu et al. (2021b) consisting of 6 stacked self- and cross-attention layers. This module outputs multi-view-aware features $\left\{\boldsymbol{F}^i\right\}_{i=1}^{N}$, where $\boldsymbol{F}^i \in \mathbb{R}^{\frac{H}{s} \times \frac{W}{s} \times C}$.

We further adopt the plane-sweep stereo technique Collins (1996); Xu et al. (2023) to construct geometric consistency. We uniformly sample $D$ candidate depths between near and far bounds. Given reference view $i$ and source view $j$, we warp features $\boldsymbol{F}_j$ to view $i$ at each depth $d_m$, resulting in $\left\{\boldsymbol{F}_{d_m}^{j \to i}\right\}_{m=1}^{D}$. These warped volumes are compared to $\boldsymbol{F}_i$ via dot-product similarity to construct a cost volume $\boldsymbol{C}^i \in \mathbb{R}^{\frac{H}{s} \times \frac{W}{s} \times D}$.

**Single-view Branch.**  We utilize the ViT backbone from Depth Anything V2 model Yang et al. (2024) to extract monocular features. The output has a spatial resolution of $1/14$ relative to the original image and is bilinearly upsampled to match the cost volume resolution, yielding monocular features $\boldsymbol{F}_{\mathrm{m}}^i \in \mathbb{R}^{\frac{H}{s} \times \frac{W}{s} \times C_{\mathrm{m}}}$.

**U-Net and Depth Prediction.**  The monocular and multi-view features $\boldsymbol{F}_{\mathrm{m}}^i$ and $\boldsymbol{C}^i$ are concatenated along the channel dimension and processed by a 2D U-Net to produce depth candidates $\boldsymbol{D}^i \in \mathbb{R}^{\frac{H}{s} \times \frac{W}{s} \times D}$. A softmax operation is applied over the depth axis, followed by a weighted summation to generate the predicted depth map.

To enhance depth quality, we employ a hierarchical cascade structure Gu et al. (2020), refining the predicted depth to $\boldsymbol{D}_{ds}^i \in \mathbb{R}^{\frac{2H}{s} \times \frac{2W}{s}}$, which is subsequently upsampled to full resolution using a DPT head Ranftl et al. (2021).

**Attribute Prediction.**  The predicted depth is used to reconstruct Gaussian positions. For estimating the remaining Gaussian attributes—such as scale multipliers, high-frequency SH coefficients, and opacity—we apply an additional DPT head, conditioned on a concatenation of the input image, predicted depth, and encoder features.

**Hyperparameter Selection.**  The downsample scale $s$ is set to 4. Channel number $C$ is set to 128, channel number $D$ is set to 128. The channel number $C_{\mathrm{m}}$ of the monocular feature is set to 64 for small model, 96 for base model, 128 for large model.

**Note:** Our implementation is consistent with DepthSplat Xu et al. (2024b) for reproducibility. No architectural modifications are made to the encoder unless otherwise stated.

## A.2  HYPERPARAMETER SENSITIVITY.

We further investigate the influence of the hyperparameters $\tau_{\mathrm{opa}}$ and $\tau_\alpha$ in Table 6. Our results indicate that SurfSplat is robust to the exact threshold values, maintaining strong performance as long as the thresholds remain within a reasonable range. This demonstrates the stability and generality of the forced alpha blending technique.

Table 6: **Ablations study on hyperparameter sensitivity.**

| Method | 256×256 (**Standard**) | | | 512×512 (**HRRC**) | | | 1024×1024 (**HRRC**) | | | Average | | |
|---|---|---|---|---|---|---|---|---|---|---|---|---|
| | PSNR↑ | SSIM↑ | LPIPS↓ | PSNR↑ | SSIM↑ | LPIPS↓ | PSNR↑ | SSIM↑ | LPIPS↓ | PSNR↑ | SSIM↑ | LPIPS↓ |
| Ours | **27.001** | **0.883** | **0.118** | **25.989** | **0.860** | **0.223** | **24.535** | **0.835** | **0.325** | **25.842** | **0.859** | **0.222** |
| $\tau_\alpha = 0.3$ | 26.921 | 0.881 | 0.120 | 25.930 | 0.860 | 0.222 | 24.816 | 0.843 | 0.317 | 25.889 | 0.861 | 0.220 |
| $\tau_{\mathrm{opa}} = 0.4$ | 26.992 | 0.883 | 0.118 | 25.957 | 0.860 | 0.222 | 24.538 | 0.835 | 0.323 | 25.829 | 0.859 | 0.221 |

## A.3 EXTENDED RESULTS AT HIGHER RESOLUTION

To further demonstrate the scalability and generalization capability of our model, we train and evaluate an extended version at higher input resolution ($256 \times 448$).

Quantitative results are summarized in Table 7, showing consistent improvements across standard and high-resolution metrics. We also visualize the rendered images and depth maps at multiple output resolutions ($\times 1$, $\times 2$, and $\times 4$) in Figure 7, Figure 8 and Figure 9, highlighting the enhanced geometric detail and texture fidelity enabled by the higher-resolution input.

Table 7: Quantitative performance of the high-resolution model.

| Method | $256 \times 448$(**Standard**) | | | $512 \times 896$(**HRRC**) | | | $1024 \times 1792$(**HRRC**) | | | Average | | |
| | PSNR↑ | SSIM↑ | LPIPS↓ | PSNR↑ | SSIM↑ | LPIPS↓ | PSNR↑ | SSIM↑ | LPIPS↓ | PSNR↑ | SSIM↑ | LPIPS↓ |
| --- | --- | --- | --- | --- | --- | --- | --- | --- | --- | --- | --- | --- |
| Ours-B | 26.190 | 0.871 | 0.134 | 25.553 | 0.861 | 0.234 | 24.197 | 0.842 | 0.329 | 25.313 | 0.858 | 0.232 |

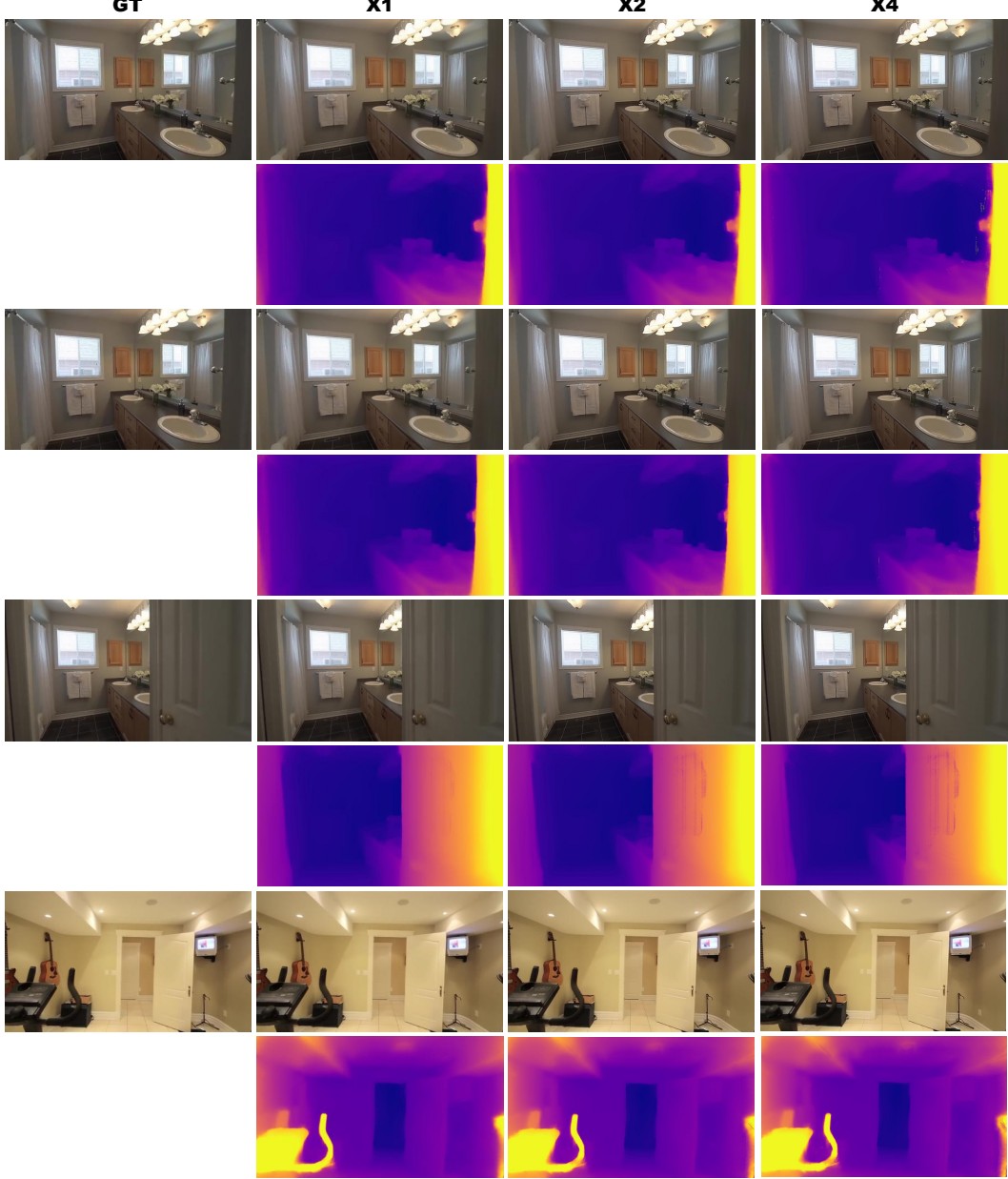

Figure 7: **Visualization of the high-resolution model.** We present rendering results (image and depth) at multiple output resolutions. As the resolution increases, our model preserves coherent geometry and appearance, revealing finer details of the scene.

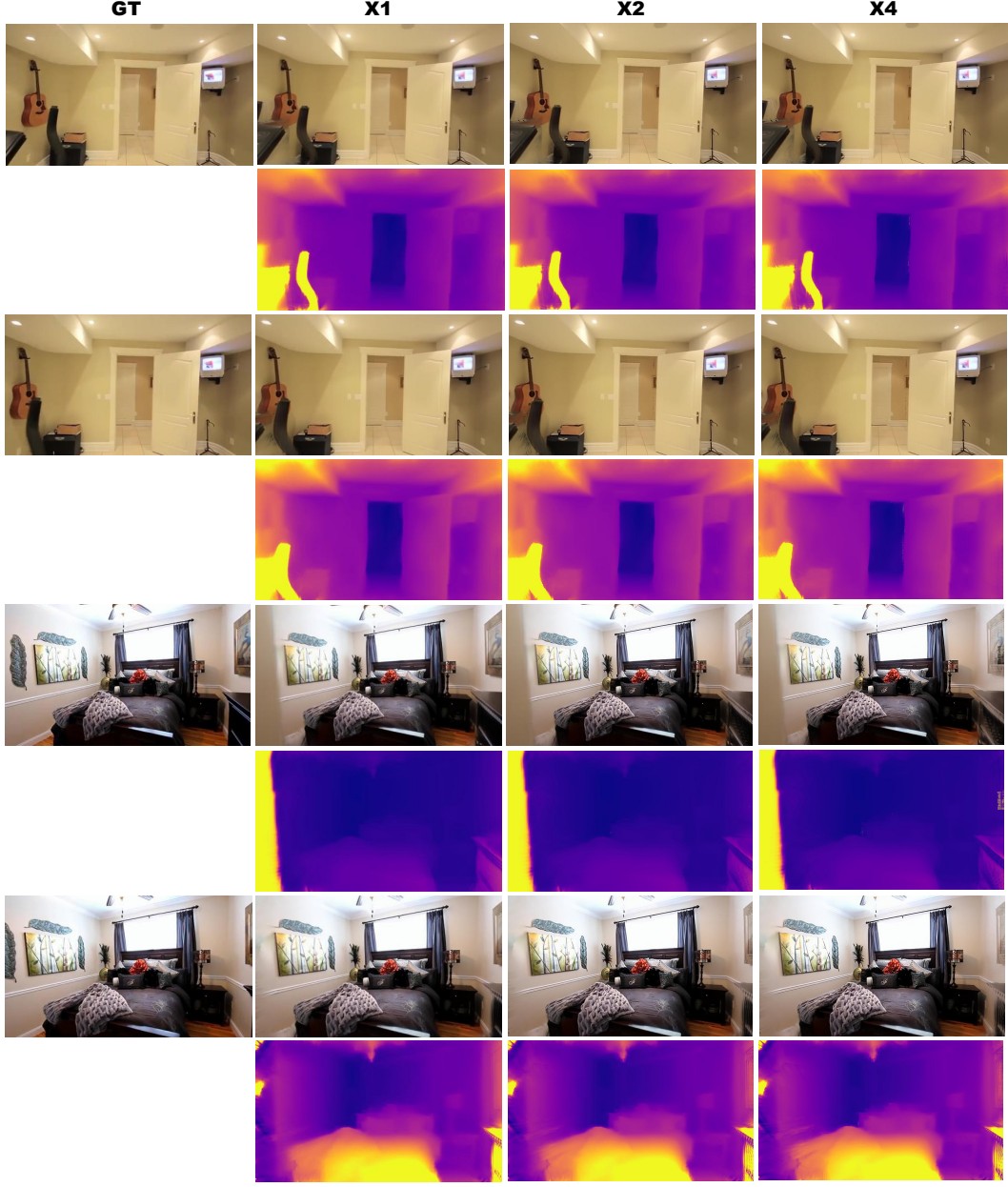

Figure 8: **Visualization of the high-resolution model.** We present rendering results (image and depth) at multiple output resolutions. As the resolution increases, our model preserves coherent geometry and appearance, revealing finer details of the scene.

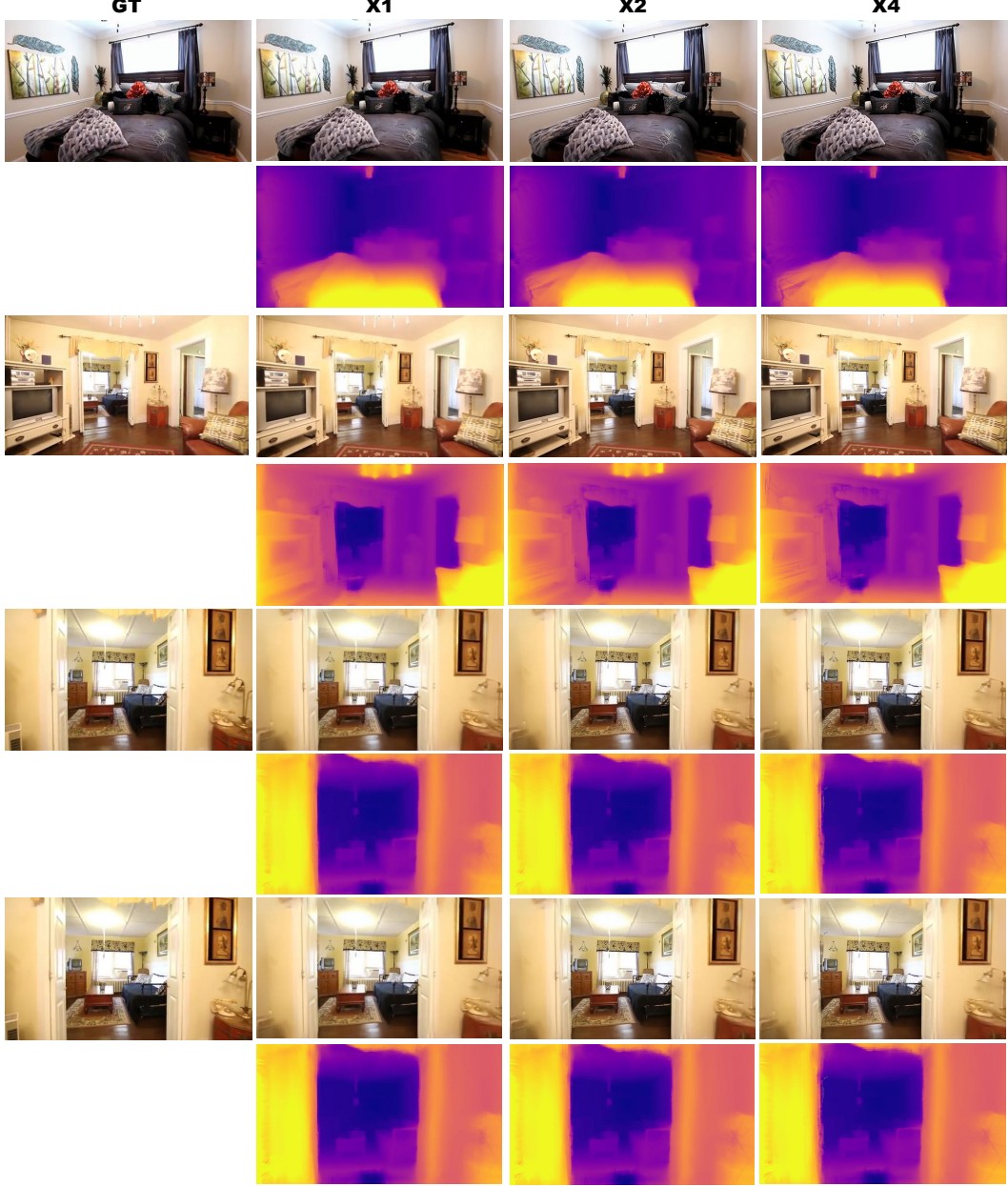

Figure 9: **Visualization of the high-resolution model.** We present rendering results (image and depth) at multiple output resolutions. As the resolution increases, our model preserves coherent geometry and appearance, revealing finer details of the scene.

