# OpenReview forum: "SurfSplat: Conquering Feedforward 2D Gaussian Splatting with Surface Continuity Priors"
_ICLR.cc/2026/Conference — ICLR 2026 Poster_

### Official Review · Reviewer_6ADx · 2025-10-29

**Soundness:** 3
**Presentation:** 3
**Contribution:** 3
**Rating:** 8
**Confidence:** 4

**Summary:**

SurfSplat is a novel method for feed-forward scene reconstruction from sparse posed-input images. Unlike previous works that employ 3D Gaussian Splats (GS) reconstructions, SurfSplat proposes a 2DGS approach. This method generates a 2DGS per pixel and incorporates two targeted additions to achieve this. The first one aligns the normals based on the local neighborhood in screen space, ensuring Surface Continuity. The second employs an enforced alpha blending technique that enhances the contribution of occluded Gaussians in the final result. This combination leads to a cleaner reconstruction where the occluded Gaussians remain optimized and contribute to the overall reconstruction quality.
For the reconstruction process, the authors extract features using two ViTs: one for single views and the other for aggregated multi-view information. These concatenated features are then transformed into 2DGS parameters using a 2D U-Net. Additionally, the authors propose a novel metric called HRRC, which renders the scene in higher resolution and compares it with upscaled ground truth images. This comparison highlights any inconsistencies or holes in the reconstructed surfaces.

**Strengths:**

- The additions to extend feed-forward-based reconstruction methods to 2DGS are well-motivated and explained.
- Overall, the paper is easy to read and follow.
- Utilizing screen-space information to construct a local tangent frame for continuous alignment of the 2DGS based on its neighborhood is a sound strategy. The implementation is also well explained.
- The forced alpha blending is particularly interesting in the feed-forward reconstruction scenario, where each pixel generates a GS that overparametrizes the scene.
- The proposed HRRC metric is intuitively comprehensible and effectively identifies critical issues like holes in the reconstruction.

**Weaknesses:**

- Although the HRRC metric is intuitively comprehensible, it would be beneficial to also demonstrate the surface normals from the 2DGS. This would not only highlight the smoothness but also serve as an indicator of the Surface Continuity Prior’s effectiveness.
- Furthermore, it would be beneficial to run the method on synthetic scenes and report F1 scores or normal alignment metrics. These metrics would provide insights into the accuracy of the geometry prediction.

**Questions:**

- Given that 2DGS are frequently easier to mesh, how do the scenes appear after meshing?
- Does the surface continuity prior use a technique to disable or weaken it for discontinuities? How are the 2DGS behaving in this areas?

---

> ### Author Response · Authors · 2025-11-22
> **Response to Reviewer 6ADx**
>
> **Response to W1:**
> Thank you for the suggestion. Our method naturally predicts a surface orientation for each 2DGS, and we have generated the corresponding normal visualizations. We provide the comparison with DepthSplat’s normals in **Figure 6**. These results highlight the improved smoothness and coherence induced by the Surface Continuity Prior and offer an intuitive indication of its effectiveness.
>
> **Response to W2&Q1:**
> Thank you for the suggestion. We agree that F1 scores and normal-alignment metrics are informative for evaluating geometry in many settings. However, we believe these metrics are not well suited for the feedforward setting considered in our work for several reasons:
>
> 1. **Feedforward models reconstruct only a small visible subset of the scene.**
>     F1 scores compare the *complete* reconstructed point cloud against the *full* ground-truth geometry. Because feedforward models typically take only 1–2 input frames, they can reconstruct only a limited visible region of the scene. As a result, the F1 score is dominated by a large number of unobserved ground-truth points, causing **all feedforward methods to obtain uniformly low scores**, even when some of them produce significantly better local geometry in the observed regions.
>
> 2. **Scale ambiguity makes point-based metrics unreliable.**
>     Feedforward models predict geometry **up to an unknown scale**, which is inherent to few-shot reconstruction. Point-based metrics such as F1 require scale-aligned geometry, so the raw predictions are not directly comparable without external alignment. This additional source of misalignment further distorts both the F1 scores and the normal-alignment metrics, making it difficult to draw meaningful conclusions about relative performance.
>
> **These limitations are precisely what motivated the introduction of HRRC in our work**. HRRC evaluates feedforward methods directly in the **image domain**, where the supervision is well defined and does not require ground-truth geometry, scale alignment, or full-scene visibility. We therefore consider HRRC a more appropriate metric for fairly comparing feedforward systems.
>
> Although full-scene normal metrics are not directly applicable in this setting, we do **generate reconstructed meshes and normal maps and compare them qualitatively with DepthSplat**. The results shown in **Figure 6** indicate that our SurfSplat produces visibly smoother and more consistent geometry, highlighting the improved geometric coherence induced by the proposed surface continuity prior.
>
> **Response to Q2:**
> We do not explicitly disable or weaken the Surface Continuity Prior in regions of discontinuity. We have experimented with traditional approaches, such as **masking areas where the depth gradient is large**, which indeed produced cleaner 2DGS mesh files locally. However, this approach had a clear drawback: the masked regions turned into holes in the reconstruction, which degraded quantitative performance.
> In practice, we found that allowing the pretrained model to operate on the full image leads to better overall results. In discontinuity regions, the predicted 2DGS typically exhibits low opacity, so these areas become effectively transparent and contribute little to the final rendering. This behavior avoids manual discontinuity handling while preserving both visual quality and metric performance.

---

### Official Review · Reviewer_VQdd · 2025-10-30

**Soundness:** 3
**Presentation:** 3
**Contribution:** 3
**Rating:** 6
**Confidence:** 4

**Summary:**

This paper targets the common failure mode of feedforward 3D reconstruction from sparse-view inputs, where existing approaches (e.g., 3DGS-based pipelines) tend to produce discontinuous, fragmented surfaces that exhibit conspicuous artifacts under close-up inspection.

Core contributions:
1、SurfSplat framework. The authors advocate using 2D Gaussian primitives (2DGS) as surface-oriented rendering elements better suited to representing continuous manifolds.
2、Surface Continuity Prior (SCP). The key innovation. Instead of letting the network blindly regress each Gaussian’s rotation and scale, SCP explicitly derives orientation and anisotropic extent from local surface normals computed on neighborhood 3D points. This imposes a geometric prior that enforces local surface continuity and smoothness of the reconstructed geometry.
3、Forced Alpha Blending (FAB). A training strategy that caps per-primitive opacity, preventing foreground splats from saturating alpha and completely occluding background layers. By ensuring gradient flow to deeper/occluded structures, FAB mitigates training instabilities and geometric misalignment.
4、HRRC metric. A new evaluation metric—High-Resolution Rendering Consistency (HRRC)—that reveals holes, cracks, and other geometric defects which standard image-space metrics often miss, by evaluating renderings at higher magnification.

Summary. By combining a strong geometric prior (SCP) with a stabilization strategy for compositing (FAB), the paper substantially improves the geometric fidelity and surface continuity of feedforward reconstruction models, and demonstrates the resulting advantages using the proposed HRRC metric.

**Strengths:**

1、High Coherence between Representation Choice and Prior (2DGS + SCP): The paper appropriately pivots to 2DGS as fundamental "surfel" elements to address the degradation issues of 3DGS under sparse views. Its core contribution lies in using the Surface Continuity Prior (SCP) to explicitly derive the rotation and scale of each primitive from local geometry, shifting the paradigm from "blind learning" to "geometrically-constrained learning." This effectively enhances the continuity and realism of the generated surfaces.

2、Proposal of a Critical and Practical Evaluation Metric (HRRC): The paper contributes the HRRC metric, which simulates a "close-up inspection" by rendering at higher resolutions. This method effectively exposes hidden geometric artifacts (e.g., holes and discontinuities) that standard metrics fail to capture. This provides strong support for the paper's claims and offers a valuable new tool for the community to assess the quality of sparse-view reconstruction.

3、Pragmatic Integration of a Dual-Branch Encoder (Monocular Prior + Multi-View Geometry): The architecture is pragmatically designed, leveraging the strong monocular prior from Depth Anything V2 and the multi-view consistency from plane-sweep cost volumes. This clear division of responsibility—monocular for completing missing textures and multi-view for ensuring consistency—places the model on a level playing field with current SOTA methods (e.g., MVSplat/DepthSplat series), ensuring a fair and effective comparison.

4、Clarity in Writing and Structure: The paper is well-structured and clearly written, with a logical flow that is easy to follow. The authors articulate the problem, the proposed solution, and its motivation, allowing readers to quickly grasp the core contributions and technical details, which is a significant merit for a high-impact publication.

**Weaknesses:**

1、Innovation Granularity Focused on Regularization and Priors; Lacks Theoretical Depth: The architecture (2DGS, dual-branch encoder) largely follows recent work. The core novelty is concentrated in the SCP+FAB+HRRC components. While the engineering results are impressive, the theoretical analysis of the SCP prior remains empirical (e.g., regarding its boundaries for curvature, texture frequency, or noise). Furthermore, a systematic comparison against alternative designs, such as "direct regression + regularization" (e.g., curvature smoothing, normal consistency loss), is notably absent.

**Questions:**

1、Potential Bias in the Proposed HRRC Metric: The HRRC metric compares the high-resolution rendered output against a ground truth image upsampled via bicubic interpolation. Since bicubic interpolation is inherently a smoothing filter that blurs fine textures and sharp edges, could this introduce a bias? Specifically, might this metric favor models that produce smoother, slightly blurred results (closer to the interpolated GT) over models that generate sharp but slightly imperfect details (which might be closer to the true high-frequency information)?

2、Weak Theoretical Support for the Forced Alpha Blending (FAB) Strategy: While FAB is shown to be effective, it essentially functions as a direct "engineering trick." The strategy of hard-clipping opacity and normalizing the final color lacks sufficient theoretical justification. This approach may introduce new rendering biases, particularly when handling semi-transparent objects or complex lighting, raising concerns about its robustness.

---

> ### Author Response · Authors · 2025-11-22
> **Response to Reviewer VQdd**
>
> **Response to W1:**
> Thank you for these thoughtful comments. We address the two aspects separately below.
>
> 1. **Theoretical motivation of SCP**
>    The SCP prior is not intended as an empirical observation, but is motivated by a basic and widely used assumption about real-world geometry: **surfaces are locally continuous**. This assumption underpins many classical 3D representations such as meshes, point-based surface models. Our formulation follows this principle by encouraging Gaussian attributes to vary consistently with the underlying local surface structure.
>
>    We acknowledge that extending SCP to handle more complex conditions such high curvature, fine textures, or heavy noise is an interesting direction for future work, and we will clarify this perspective more explicitly  as an open extension in the revised paper.
>
> 2. **Comparison with “direct regression + regularization” approaches**
> Alternative designs such as direct regression coupled with curvature-smoothing or normal-consistency losses similarly aim to enforce local surface coherence. Our approach pursues the same geometric objective but does so **within the forward prediction process**, by constraining Gaussian attributes through the inferred surface geometry rather than adding separate regularization losses. In practice, SCP directly constrains the predicted Gaussian attributes based on the underlying surface geometry, thereby already achieving the same objective as these regularizers.
>
> **Response to Q1:**
>
> Thank you for raising this concern about potential bias in the HRRC metric. We address it from both an empirical and a conceptual perspective.
>
> 1. **Empirical Validation on Native High-Resolution Data**
>    To examine whether HRRC might unfairly favor smoother reconstructions, we conducted additional experiments on the high-resolution version of the **DL3DV** dataset. We randomly sampled a representative subset for evaluation and ensured that all methods were tested under identical conditions. The results are reported in  **Table 7**.
>    In these experiments, the **relative performance rankings remained fully consistent with those observed under HRRC evaluation**, even without any bicubic upsampling. This suggests that the conclusions drawn using HRRC largely transfer to native high-resolution evaluations and that the metric does not systematically amplify the advantages of our method.
>
> 2. **Conceptual Justification of HRRC as a Fair Evaluation Metric**
>    The reviewer raised a concern that HRRC might favor overly smooth reconstructions. However, our experiments suggest that the performance degradation of competing methods at high resolution is mainly due to **missing regions in the reconstruction**, as also reflected in their generated depth maps, rather than to the presence or absence of sharp features. This is in line with our motivation: the proposed regularizations are designed to reduce structural discontinuities, which are common failure modes in Gaussian-based feedforward pipelines. Consequently, HRRC tends to capture differences in reconstruction reliability, rather than merely rewarding smoothness.
>
> **Response to Q2:**
>
> We agree that the Forced Alpha Blending (FAB) is primarily a practical solution, and we do not claim a strong theoretical foundation for it. Nevertheless, our experiments empirically validates that it consistently improves reconstruction quality and reduces typical failure cases across several datasets. Its motivation is straightforward: feedforward Gaussian pipelines in our setting operate predominantly in the **opaque surface domain**, where unstable opacity predictions can lead to holes and inconsistent transmittance. FAB directly addresses this by stabilizing opacity and normalizing color accumulation, which empirically reduces these failure cases.
>
> It is noteworthy that our design of FAB is explicitly tailored to **solid surfaces**, which matches the target domain and benchmarks considered in our paper. We recognize that semi-transparent materials and complex lighting configurations raise additional challenges and may indeed require different or extended treatments. However, these aspects are outside the scope of our current work and we will make the scope of our paper clearer in the revised version.

---

### Official Review · Reviewer_pqgf · 2025-10-30

**Soundness:** 3
**Presentation:** 3
**Contribution:** 2
**Rating:** 6
**Confidence:** 3

**Summary:**

This paper introduces SurfSplat, a feedforward framework that replaces 3D Gaussian splatting (3DGS) with 2D Gaussian splatting (2DGS) to improve geometry reconstruction and texture fidelity in sparse-view 3D scene reconstruction. The key ideas include a surface-continuity prior linking splat shape and orientation to surface normals, and a forced-alpha blending strategy to prevent opacity collapse. The paper also proposes High-Resolution Rendering Consistency (HRRC) as a new metric to evaluate fine-grained artifacts. Experiments on RealEstate10K, DL3DV, and ScanNet show consistent improvements over baseline methods.

**Strengths:**

1. Solid methodological design:
Using 2D splats (surfels) instead of 3D Gaussians is an interesting design choice that simplifies the rendering pipeline while retaining expressive power. The authors carefully justify why this helps preserve anisotropy and continuity.
2. Comprehensive experiments:
Results on multiple datasets (RealEstate10K, DL3DV, ScanNet) demonstrate robustness and consistent improvements across settings. Qualitative visualizations effectively support the claims.

**Weaknesses:**

1. Limited theoretical grounding for the continuity prior:
While the prior is intuitive, its mathematical justification remains somewhat heuristic. The authors could discuss more explicitly how the coupling of rotation and scale influences convergence and stability.
2. Ablation analysis could be deeper:
The ablations are primarily qualitative. Quantitative ablations showing how much each component (surface prior, alpha blending, HRRC) contributes to the final performance would strengthen the empirical evidence.

**Questions:**

1. Theoretical Motivation of the Surface-Continuity Prior

Could the authors clarify the theoretical reasoning for coupling the splat’s rotation and scale with local surface orientation? Is this prior based on any assumption of local planarity or normal consistency, and how sensitive is the approach to errors in predicted normals?

2. Quantitative Validation and Ablation

Beyond qualitative comparisons, could the authors provide quantitative ablations that isolate the contributions of the surface-continuity prior and forced-alpha blending? For example, what are the PSNR / SSIM / HRRC changes when each component is removed?

---

> ### Author Response · Authors · 2025-11-22
> **Response to Reviewer pqgf**
>
> **Response to W1&Q1:**
> Thank you for ointing out the need for a clearer theoretical motivation. We clarify the intent and role of the Surface Continuity Prior (SCP) below.
>
> Our **Surface Continuity Prior (SCP)** is motivated by a simple and widely used observation: feedforward Gaussian-based methods ultimately aim to reconstruct **continuous 3D surfaces**. Since each pixel corresponds to the projection of a point on such a surface, enforcing local consistency among predicted Gaussian attributes can be viewed as more than a purely heuristic choice. It follows from the assumed underlying surface geometry. The prior is instantiated using the predicted **depth map**, which provides sufficient local geometric cues to guide this coupling.  Our experiments across multiple datasets show that the method is **robust to normal or depth errors**. At true discontinuities, the predicted opacity tends to be low, so these regions have limited influence on the final rendering.
>
> More specifically, the coupling between **rotation and scale** arises because these parameters **jointly describe the local tangent patch represented by each Gaussian**. Rotation determines the orientation of the patch, while scale determines its spatial support. When these two attributes vary incoherently across neighboring pixels, optimization can become less stable and may lead to missing regions or fragmented surface structures. By encouraging rotation and scale to vary smoothly in a correlated manner, SCP helps reduce this ambiguity and tends to stabilize optimization in practice.
>
> As detailed in the paper Section 3.3, we provide derivations that make this projection relationship explicit, giving SCP a more principled mathematical form than an ad-hoc smoothing term. We would be happy to further clarify the specific aspect of the rotation–scale interaction that requires deeper explanation.
>
> **Response to W2&Q2:**
> Thank you for this suggestion. We would like to clarify that our ablation analysis already includes both **quantitative** and **qualitative** evaluations. We agree that their role could be made more explicit and we will revise our paper to better highlight this.
>
> - **Quantitative ablations:**
> **Tables 4 and 5 in** report numerical results for different model variants in terms of PSNR / SSIM / HRRC., following the same order in which the modules are introduced. This setup allows us to examine the contribution of:
>   1. the **Surface Continuity Prior (SCP)**,
>   2. the **Forced Alpha Blending (FAB)** strategy, and
>   3. the final full model.
>
> - **Qualitative ablations:**
>   **Figure 5** complements these tables by visually illustrating typical failure cases and the corresponding improvements brought by each module.
>
>   Our ablation protocol follows the design sequence we adopted for the method:
>
>   1. SCP is added first, which improves continuity but may introduce visible surface deformations;
>   2. FAB is then introduced to alleviate this optimization issue;
>   3. the complete model combines SCP and FAB.
>
> Regarding the role of HRRC:
> HRRC is  **only used as an evaluation metric**, never as a training objective. The training losses are explicitly listed in **Section 3.5**. Therefore, HRRC does not influence the learning process, and we do not expect it to confound the ablation analysis. We will clarify this point more clearly in the revised manuscript.
>
> If there is a specific additional quantitative breakdown the reviewer is looking for, we are happy to provide further clarification.

---

> > ### Comment · Reviewer_pqgf · 2025-11-27
> >
> > Thank you to the authors for addressing my concerns; I maintain my original score.

---

### Official Review · Reviewer_3CPe · 2025-11-01

**Soundness:** 3
**Presentation:** 3
**Contribution:** 3
**Rating:** 6
**Confidence:** 4

**Summary:**

This paper presents a feedforward 3D reconstruction framework from sparse views based on 2D Gaussian Splatting (2DGS), as an alternative to the more common 3D Gaussian Splatting (3DGS) frameworks. Existing feedforward approaches, such as PixelSplat and MVSplat, which use 3D Gaussians as primitives, often suffer from discontinuities (holes) in the reconstructed surfaces and color-biased artifacts when given sparse-view inputs. To address these issues, this paper introduces a surface continuity regularization that enforces smoothness among Gaussians within a local 3D neighborhood. Additionally, a forced alpha blending strategy is proposed to mitigate opacity saturation during rendering. To further highlight the shortcomings of current evaluation protocols, the paper also introduces a new metric, High-Resolution Rendering Consistency (HRRC), designed to better assess geometry fidelity. Experiments on RealEstate10K, DL3DV, and ScanNet demonstrate that the proposed method achieves superior geometric and visual fidelity compared to recent feedforward baselines, especially under the HRRC evaluation.

**Strengths:**

* Although the overall architecture of predicting per-pixel gaussian is not new, the proposed strategy of enforcing surface smoothness and forced alpha blending is relatively new and interesting.

* The motivation of the paper is clear, and the paper is well-written and the proposed method is easy to follow and the overall presentation is clear.

* The paper is solving a relevant problem. Reconstructing a high fidelity surface from sparse input is a relevant problem.

* Comprehensive evaluation is done on various datasets to show the effectiveness of the method.

**Weaknesses:**

* Why use a single Gaussian per pixel? While the method performs well against baselines that also use a single Gaussian per pixel on the novel view synthesis task (both within and cross-dataset), this design seems to limit performance compared to methods like PixelSplat and HiSplat, which use multiple Gaussians per pixel. Did the authors experiment with predicting multiple Gaussians per pixel? From Tables 1 and 2, it appears that doing so might address the observed issues more effectively than relying solely on the proposed surface continuity and forced alpha blending techniques.


* HRRC may favor the proposed regularization strategies. While HRRC appears to be an effective metric for evaluating feedforward GS-based reconstruction frameworks, it might inherently bias the evaluation toward smoother reconstructions due to the use of upsampled ground-truth images (via bicubic interpolation). Have the authors considered training the model on low-resolution input images and then evaluating on the original ground-truth resolution? Would such an evaluation be valid or provide a fairer measure of reconstruction fidelity? A small experiment with this setting will be helpful.

**Questions:**

Please refer to weakness.

Minor issues:
* Line 193: Insufficien -> Insufficient

* Line 162: the notation of camera intrinsics and poses is not clear.

---

> ### Author Response · Authors · 2025-11-22
> **Response to Reviewer 3CPe**
>
> **Response to W1:**
> Thank you for your insightful question. Our choice to use a **single Gaussian per pixel** follows a core design principle: **pursuing the most efficient possible 3D representation while maintaining strong visual performance**.
>
> While recent feedforward methods often prioritize visual quality and multiple Gaussians per pixel can indeed improve appearance, this design choice comes with several important drawbacks:
>
> 1. **Computational and Memory Inefficiency**
>
>    Increasing the number of Gaussians per pixel leads to significantly heavier computation and slower rendering, especially at higher resolutions or when processing many input views. Moreover, the uncontrolled growth in the number of primitives can easily cause out-of-memory (OOM) issues.
>
> 2. **Storage and Transmission Overhead**
>
>    Representations with large numbers of primitives require substantially more storage, which complicates model deployment, compression, and transmission, particularly in real-time or resource-constrained settings.
>
> 3. **Negative Impact on Downstream Tasks**
>
>    Many downstream systems depend on compact and efficient intermediate 3D representations. Inflating the number of Gaussians per pixel can introduce bottlenecks in such pipelines, harming both efficiency and overall performance.
>
> Instead of increasing the primitive count, our method **addresses these core challenges without introducing extra Gaussians**. We focus on enforcing **surface continuity** and **forced alpha blending**, which directly mitigate the key issues while preserving efficiency. As shown in **Tables 1 and 2**, these components enable strong performance within a single-Gaussian framework.
>
> **Response to W2:**
>
> Thank you for raising this important concern about potential bias in the HRRC metric. We address the question from both an empirical and a conceptual standpoint.
>
> 1. **Empirical Validation on Native High-Resolution Data**
>    To assess whether HRRC might unfairly favor smoother reconstructions, we conducted additional experiments on the high-resolution version of the **DL3DV** dataset. We randomly sampled a representative subset for evaluation and ensured that all methods were tested under identical conditions. The results are reported in **Table 7**.
>    Across these experiments, the **relative performance rankings remained fully consistent with those observed under HRRC evaluation**, even without any bicubic upsampling. This indicates that the conclusions drawn from HRRC reliably transfer to native high-resolution evaluations and that the metric does not artificially amplify the advantages of our method.
>
> 2. **Conceptual Justification of HRRC as a Fair Evaluation Metric**
>    The reviewer raised a concern that HRRC might favor overly smooth reconstructions. However, our experiments indicate that the performance degradation of competing methods at high resolution is primarily caused by **missing regions in the reconstruction**, as also evidenced by their generated depth maps, rather than by the presence or absence of sharp features. This is consistent with our motivation: the proposed regularizations are specifically designed to reduce structural discontinuities, which are fundamental failure modes in Gaussian-based feedforward pipelines. As a result, HRRC captures meaningful differences in reconstruction reliability rather than merely rewarding smoothness.
>
> **Response to Question:**
> Thank you for pointing out the typo and we have revised the paper accordingly.

---

### Author Response · Authors · 2025-11-22
**General Responses to All Reviewers**

We sincerely thank all reviewers for their time and constructive feedback.

In response to the comments, we have revised several sentences throughout the paper to improve clarity, with all changes highlighted in red in the revised paper. We summarize the main experimental additions and revisions below:

1. Added a quantitative comparison on native high-resolution dataset to demonstrate the effectiveness of HRRC in **Table 7**.
2. Added a qualitative comparison of reconstructed mesh and surface normal with DepthSplat in **Figure 6**.

We would be pleased to provide further clarifications or additional analyses if the reviewers have any remaining questions.

---

### Meta-Review · Area_Chair_WiPw · 2026-01-06

**Summary:**

The paper presents a generalizable 3D reconstruction framework built upon the 2D Gaussian Splatting. To address limitations in existing methods, the authors introduce a surface continuity prior and a forced alpha blending strategy to enhance reconstruction quality. The reviewers acknowledged the novelty and strong empirical results, leading to initial scores of (6, 6, 6, 8). Concerns regarding the theoretical grounding and potential bias of the new HRRC metric were effectively addressed through a robust rebuttal and additional quantitative experiments. I recommend the paper for Acceptance.

**Reviewer Concerns:**

Two major concerns have been addressed in the rebuttal:
- HRRC metric: to address the concerns on metric bias, the authors introduced Table 7, utilizing native high-resolution data from the DL3DV dataset. This quantitative comparison successfully demonstrated the effectiveness and reliability of the HRRC metric.
- Geometric fidelity: the authors provided a qualitative comparison of reconstructed meshes and surface normals against DepthSplat in Figure 6. These visualizations demonstrated superior geometric fidelity.

**Reviewer Scores:**

The manuscript received initial scores of (6, 6, 6, 8). During the discussion phase, one reviewer explicitly confirmed they would maintain their positive rating, while the remaining three reviewers did not provide further comments. From my experience, as the rebuttal addressed the technical concerns regarding metric bias and geometric accuracy, I would approximate the final consensus score to be between 6.5-7.0, leading to a clear recommendation for Acceptance.

---

### Decision · Program_Chairs · 2026-01-26

Accept (Poster)